# Generating Realistic Synthetic Relational Data through Graph Variational Autoencoders

**Ciro A. Mami***
Aindo
Fintech District, Milan, Italy
ciroantonio@aindo.com

**Andrea Coser**
Aindo
AREA Science Park, Trieste, Italy
andrea@aindo.com

**Eric Medvet**
University of Trieste
Trieste, Italy
emedvet@units.it

**Alexander T.P. Boudewijn**
Aindo;
Katholieke Universiteit Leuven
Heverlee, Belgium
alexander@aindo.com

**Marco Volpe**
Business Integration Partners (BIP)
Milan, Italy
marco.volpe@mail-bip.com

**Michael Whitworth**
Chaucer
Manchester, England, United Kingdom
michael.whitworth@chaucer.com

**Borut Svara**
Aindo
AREA Science Park, Trieste, Italy
b@aindo.com

**Gabriele Sgroi**
Aindo
AREA Science Park, Trieste, Italy
gabriele@aindo.com

**Daniele Panfilo**
Aindo;
University of Trieste
AREA Science Park, Trieste, Italy
daniele@aindo.com

**Sebastiano Saccani**
Aindo
AREA Science Park, Trieste, Italy
sebastiano@aindo.com

## Abstract

Synthetic data generation has recently gained widespread attention as a more reliable alternative to traditional data anonymization. The involved methods are originally developed for image synthesis. Hence, their application to the typically tabular and relational datasets from healthcare, finance and other industries is non-trivial. While substantial research has been devoted to the generation of realistic tabular datasets, the study of synthetic relational databases is still in its infancy. In this paper, we combine the variational autoencoder framework with graph neural networks to generate realistic synthetic relational databases. We then apply the obtained method to two publicly available databases in computational experiments. The results indicate that real databases' structures are accurately preserved in the resulting synthetic datasets, even for large datasets with advanced data types.

---

*Corresponding author

NeurIPS 2022 Workshop on Synthetic Data for Empowering ML Research.

# 1 Introduction

Increases in computational power as well as general awareness have brought data sciences to the forefront of modern business management. However, with the upsurge in data usage came an upsurge in privacy concerns and protocols. Since the introduction of the General Data Protection Regulation (GDPR) in the European Union, over 1,100 fines have been issued [1], a number that grows steadily.

Legal (see, e.g. [2, 3]), scientific (see, e.g., [4, 5]) and anecdotal evidence indicate that traditional anonymization techniques offer insufficient protection against data breaches. A famous example is the 2014 New York City Taxi and Limousine Commission data breach [6], in which hackers could infer drivers' addresses and incomes. Deep-learning generative models (DLGMs) have gained attention as a more reliable alternative to anonymization [7, 8, 9]. Rather than distorting existing datasets to mask its subjects, these methods create entirely synthetic new datasets. However, these synthetic datasets retain all the intricate patterns of the real datasets used in the DLGM's training.

DGLMs such as variational autoencoders (VAE) have been applied succesfully for realistic image generation [10, 11] and more recently for generating realistic tabular data with statistically independent rows [12, 13, 14, 15, 16, 17]. However, the lack of methods for more advanced database structures such as relational databases limits the use cases to which DLGMs can be applied. Some methods for relational databases with single main tables and multiple secondary tables have been proposed (see, e.g. [18]). However, these methods do not generalize to more complex database, lacking the ability to process many interconnected tables.

In this paper, we introduce graph-VAE, a method based on graph neural networks and VAE to generate realistic synthetic relational data. We also evaluate the effectiveness of the method using publicly available datasets through model compatibility metrics and a privacy assessment. Results indicate that graph-VAE generates relational databases that accurately securely preserve real databases' structures, even for large datasets with advanced data types.

# 2 Background

## 2.1 Variational autoencoders

Given a table of data $T$, the objective of a deep-learning generative model (DLGM) is to construct a new dataset $T'$ such that $T$ and $T'$ have the same distribution. A variational autoencoder (VAE) does this by connecting *encoder* and *decoder* neural networks [12, 16]. Encoder $q_\phi(z|x)$, $x \in T$, parameterized by $\phi$, maps $T$ to a probability distribution ("code"). Decoder $p_\psi(x|z)$, parameterized by $\psi$, maps *latent variables* $z$ to the data space. Training a VAE is maximizing the (log-)likelihood that code samples are taken from the original data's distribution. Structure is imposed on the encoder by minimizing its KL-divergence with a prior $P$ (typically Gaussian). This gives the loss function in expression (1), for $D_{KL}$ the KL-divergence and $\beta$ a hyperparameter.

$$\mathcal{L}(\phi, \psi) = -\sum_{x \in T} \mathbb{E}_{z \sim q_\phi(z|x)} \log p_\psi(x|z) + \beta \cdot D_{KL}(q_\phi || P) \tag{1}$$

## 2.2 Graph neural networks

Graph neural networks (GNN) are structures designed to reconcile neural networks with advances data types modeled through graph theory. GNN models typically involve a *message passing* step [19, 20]. Such a step distributes information contained in feature values of a specific vertex to its neighborhood, allowing subsequent neural network layers to infer patterns on a larger structural level than that of mere individual node features. Several layers of message passing can be cascaded.

Let $G = (V, E)$ be a graph, in which each vertex $t \in V$ has a set of *vertex features* $X_t$ and each edge $\{s, t\} \in E$ has a set of *edge features* $X_{\{s,t\}}$. Additionally, a collection $H_s^l$ of hidden vertex variables is associated with each vertex $t \in V$. Let $N(t)$ denote the neighborhood of vertex $t \in V$, that is: $N(t) := \{s \in V : \{s, t\} \in E\}$. Suppose there are $L$ message layers. In each layer $l$ and for each vertex $s \in V$, vertex features $h_s^l \in H_s^l$ are updated through a differentiable *message function* $M_l$ and a differentiable *update function* $U_l$ according to equation (2).

$$\begin{cases} m_t^{l+1} & = \sum_{s \in N(t)} M_l(h_t^l, h_s^l, X_{\{s,t\}}) \\ h_t^{l+1} & = U_l(h_t^l, m_t^{l+1}) \end{cases} \tag{2}$$

For a given vertex, the message function computes messages from adjacent vertices, which are then aggregated over the vertex' neighborhood. We aggregate through summation, though other aggregation operations may also be chosen. The update function uses the aggregated messages to update the vertex attributes.

## 3 Graph Variational Autoencoders for Realistic Synthetic Relational Data

### 3.1 Relational data

Our graph variational autoencoder (graphVAE) combines VAE with GNN to generate realistic tabular data. A *dataset* $D$ is a collection of one or more tables $\{T_1, T_2, \ldots\}$. A *table* $T$ is a collection of rows $\{t_1, \ldots, t_n\}$. A row $t$ is a tuple defined over a sequence of attributes $A(T) = (a_1, \ldots, a_p)$ and represents an *entity* in the real world: we denote by $v(t, a) \in V(T, a) \cup \emptyset$ the value of the attribute $a$ for the row $t$, with $V(T, a)$ being the *domain* of the attribute $a$ and $\emptyset$ representing the undefined value (i.e., $v(t, a) = \emptyset$ means that $a$ is missing for $t$). We say that an attribute $a$ is *unique* for a table $T$ if and only if the mapping $t_i \mapsto v(t_i, a)$ is injective for all $t_i \in T$ and $a$ is always defined in $T$.

We say that a dataset $D$ is *relational* if the following conditions hold:

1. it contains at least two tables;
2. at least one *primary table* $T^\star$ has a unique attribute $a^\star$;
3. for each other *secondary* table $T$ in the dataset, $A(T) \ni a^\star$ and $\forall t \in T, \exists t^\star \in T^\star : v(t, a^\star) = v(t^\star, a^\star)$.

Intuitively, a relational dataset is a dataset with a primary table $T^\star$ describing some entities, one row per entity, and other tables describing some other entities, each one linked to one specific entity of $T^\star$. For example, a primary table may contain information about customers, and a secondary table may contain data about specific purchases. We refer to the attribute $a^*$ as the *identifier attribute*.

### 3.2 Graph theoretic representation of relational data

To model relational databases as graphs, we follow [21]. Consider a relational dataset $D$ with a primary table $T^*$ and special attribute $a^*$. Define the *relational graph* $G(D) = (V(D), E(D))$ according to equation (3).

$$\begin{cases} V(D) & = \{t \,|\, \exists T \in D : t \in T\} \\ E(D) & = \{\{s, t\} \,|\, (s \in T^*) \wedge (\exists T \in D \setminus T^* : t \in T) \wedge (v(s, a^*) = v(t, a^*))\}, \end{cases} \tag{3}$$

so that the identifier attribute $a^*$ (unique in $T^*$) links records in secondary tables back to the specific row in the primary table with the corresponding identifier value. We note that the definition of a relational graph also applies to relational databases with multiple primary tables and identifier attributes, by computing the edge set for all identifier attributes (including nested structures where a table $T$ may serve as a primary table to a table $T'$, but as a secondary table to a table $T''$).

For a relational graph $G(D)$ of a relational dataset $D$, we define the *edge list attribute*, denoted by $a^\dagger$ for all $T \in D$ according to equation (4)

$$v(t, a^\dagger) = \{s \,|\, \{s, t\} \in E(D)\} \tag{4}$$

### 3.3 GraphVAE architecture

Let $D$ be a relational dataset with a primary table $T^*$, secondary tables $T^1, T^2, \ldots$ and relational graph $G(D) = (V(D), E(D))$. Let $\phi_\dagger$ be the mapping such that $\phi_\dagger(G(D)) = a^\dagger$, for $a^\dagger$ the edge list of $G(D)$. Use this mapping to obtain and store $a^\dagger$. Next, follow the preprocessing steps detailed in [18] for each table $T \in D$. Particularly, an invertible function $\phi_{type}$ maps tables containing numeric,

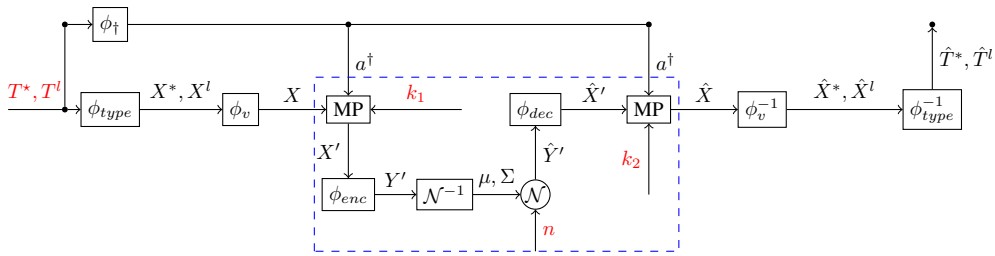

Figure 1: Schematic representation of the steps for going from a relational dataset $D = (T^\star, T^1, \ldots, T^h)$ to a compatible synthetic relational dataset $\hat{D} = (\hat{T}^\star, \hat{T}^1, \ldots, \hat{T}^h)$. In red, the parameter and inputs provided by the user; in blue dashed rectangles, the core steps of the graphVAE, other steps having to do with pre- and post-processing. Abbreviation "MP" indicates a cascade of four message passing layers.

categorical and date-time attributes to normalized tables of only numeric attributes. Subsequently, an invertible function $\phi_v$ maps all tables $T \in D$ to a single table as follows. First, add the edge list attribute $a^\dagger$ to all tables. Next, initialize a table $X$ with attribute set $A(X) = \bigcup_{T \in D} A(T) \cup \{a^\dagger\}$. Note that, as each table contains the special attribute $a^*$, we have $|A(X)| = \sum_{T \in D} A(T) - |D| + 2$. Subsequently, all rows of all tables $T \in D$ are added to $X$ through vertical concatenation, where

$$\forall a \in A(X): \quad v(x, a) = \begin{cases} v(t, a), & \text{if } a \in A(T) \cup \{a^\dagger\} \\ 0, & \text{otherwise,} \end{cases} \tag{5}$$

and $v(t, a^\dagger)$ is the adjacency list of the vertex corresponding to vertex (row of some table) $t$. The table $X$ is a representation of a graph with vertex features and adjacency lists.

Subsequently, $k_1$ message passing layers are applied to the graph represented by $X$. The number $k_1$ is a hyperparameter determined by the user. Larger values of $k_1$ result in information being more distributed throughout the graph. The message and update functions are provided in equations (6).

$$\begin{cases} M_l(h_s^l, h_t^l, x_{\{s,t\}}) &=& A(X_{\{s,t\}})h_s^l \\ U_l(h_t^l, h_t^{l+1}) &=& \text{GRU}(h_t^l, m_t^{l+1}), \end{cases} \tag{6}$$

where GRU is a gated recurrent unit (cf. [21]); $A$ is a a matrix of learned parameters that map edge features to the appropriate dimensions; and $h_t^l$ is the value $v(t, a)$ for $t \in X$ in message passing layer $l$.

After the message passing phase, attribute $a^\dagger$ is discarded and the remaining table, say $X'$, is fed into a variational autoencoder with a standard multivariate normal Gaussian prior, the dimension of which is a hyperparameter chosen by the user. Random samples $\hat{Y}'$ in the latent space are taken and transferred back to the data space by the decoder. Subsequently, a user-specified number $k_2$ of additional message passing layers are applied before the data is transferred to the original primary and secondary table structure through the inverse preprocessing functions. A standard VAE loss function is invoked during training (see equation (1)). This is illustrated in a computational diagram in Figure 1.

## 4 Experimental Results

### 4.1 Datasets

We applied our method to two publicly available datasets: the BasketballMen dataset from the Relational Dataset Repository [22] and the Rossmann Store Sales dataset from Kaggle [23]. These sets have a single primary table and two and one secondary tables, respectively. All attribute types (numeric, categorical, date-time) are represented in these sets. Both secondary tables of the BasketballMen dataset are linked to the primary dataset through the same identifier attribute. While the Rossmann dataset has only one child table, it has a large data volume and a more complex mixture of datatypes. Table 1 summarizes the salient information of the datasets.

| Dataset | $|T|$ | $|A(T)|$ | N | C | T |
|---|---|---|---|---|---|
| BasketballMen | | | | | |
| *Primary* | 5 062 | 4 | 2 | 2 | 0 |
| *Secondary 1* | 1 608 | 7 | 7 | 0 | 0 |
| *Secondary 2* | 23 751 | 6 | 5 | 1 | 0 |
| Rossmann Store | | | | | |
| *Primary* | 1 004 | 9 | 5 | 4 | 0 |
| *Secondary* | 915 223 | 8 | 2 | 5 | 1 |

Table 1: Overview of the datasets used in the experiments, one row per table. Column $|T|$ indicates the number of rows in the table. Column $|A(T)|$ indicates the number of attributes. Columns N, C, and T indicate the number of numeric, categorical, and time attributes respectively.

## 4.2 Experimental set-up

We used the architecture from Figure 1 to generate synthetic data with each of the two datasets as input. Hyperparameters are initiated based on available domain knowledge. They are then optimized manually by repeating the experiments with slightly altered values. The obtained hyperparameters are provided in Table 2.

To measure the degree to which generated synthetic data preserves the structure of the input data, we use model compatibility (MC) through the following procedure: 1) we divide the input dataset into a training set (80%) and a test set (20%); 2) We train the graphVAE using the training set; 3) We use the trained model to construct the synthetic dataset; 4) We train two gradient boosted trees (XGBoost) classifiers: one on the on the join of the primary and secondary tables of the training set; and one on the join of the synthetic primary and secondary tables. Both classifiers have the same target column based on domain knowledge: "position" (categorical, position of a player) for the basketball dataset and "Promo" (Boolean, whether or not there was a promotion in the store) for the Rossman store dataset); 5) We apply both models to the same test set (20% of the original dataset) and evaluate standard machine learning metrics (ROC AUC and F1-score) of the trained models; 6) We compare the performance metrics of the model built on the training set to those built on the test set. If the synthetic data accurately mimics the properties of the input dataset, the performances of both models should be roughly equal.

As a quantitative indicator of model compatibility, we use the metric

$$MC(D, \hat{D}) = \left| 1 - \frac{e(m, D_{test})}{e(\hat{m}, D_{test})} \right|, \tag{7}$$

where $D$ and $\hat{D}$ are the real and synthetic datasets, respectively; $m$ and $\hat{m}$ are the model built on the training and synthetic datasets, respectively; and $e$ is the effectiveness metric (ROC AUC or F1-score). The closer $MC(D, \hat{D})$ is to zero, the more similar the performance of the two models is on the test data: it indicates that patterns learned from the original dataset are also learned from the synthetic data.

To quantify the degree to which synthetic data protects the privacy of individuals contained in the original dataset, we follow [18, 24, 17] in using nearest neighbor distance-based metrics. Intuitively, these metrics measure whether synthetic records are too close to real records, making it possible to re-identify a real individual, or their characteristics.

In particular, define the *nearest neighbor distribution* $\text{NN}_{X,Y} : X \to \mathbb{R}$ of two datasets $X$ and $Y$ through

$$\text{NN}_{X,Y}(x) := \begin{cases} \min_{y \in Y \setminus \{x\}} d(x,y), & \text{if } X = Y \\ \min_{y \in Y} d(x,y), & \text{if } X \neq Y, \end{cases} \tag{8}$$

for $d$ the Euclidean distance measure. We partition the original dataset $D$ into two sets $D_1, D_2$ of the same cardinality. We then compute the fifth percentile $\alpha$ of the distribution $\text{NN}_{D_1,D_2}/\text{NN}_{D_1,D_1}$. The *privacy score* $\mathcal{P}(\hat{D})$ of synthetic dataset $\hat{D}$ is then defined as

$$\mathcal{P}(\hat{D}) := \frac{1}{|D|} \left| \left\{ x \in D : \frac{\text{NN}_{D,\hat{D}}(x)}{\text{NN}_{D,D}(x)} < \alpha \right\} \right|, \tag{9}$$

| Param | Description | Basketball | Rossmann |
|-------|-------------|------------|----------|
| $n$ | VAE latent space dimensionality | [8, 10, 10] | [8, 8] |
| $k_1$ | # message passing layers | 4 | 4 |
| $k_2$ | # message passing layers | 4 | 4 |
| $\beta$ | VAE loss terms tradeoff | [1,1,1] | [5,5,5] |

Table 2: Summary of hyperparameter values. Vectors provide separate parameter values for the primary table in the first entry, followed by those of the secondary table(s).

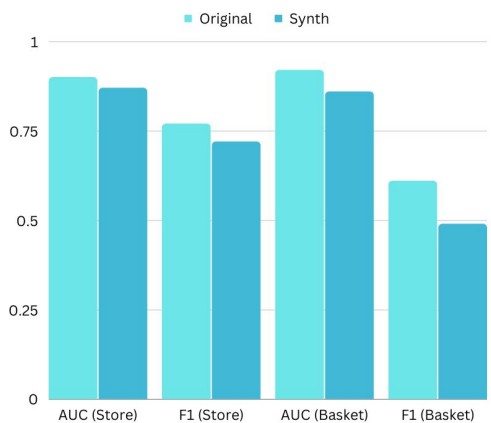

Figure 2: Effectiveness metrics evaluated on training (light blue) and synthetic data (dark blue)

with synthetic dataset $\hat{D}$ considered sufficiently private if $\mathcal{P}(\hat{D}) \leq 0.05$ and insufficiently private otherwise. Intuitively, the metric assesses the proportion of original records that are closer to synthetic records, than the 5% of smallest distances between records from $D_1$ and $D_2$.

### 4.3 Results and discussion

The model compatibilitiies for the BasketballMen dataset are 0.07 (ROC AUC) and 0.24 (F1-score). For the Rossmann Store Sales dataset, they are $0.03$ (ROC AUC) and $0.07$ (F1-score). Figure 2 shows the performance metrics of the models evaluated on the test dataset. In both experiments, the value $\mathcal{P}(\hat{D})$ is significantly below $0.05$ (BasketballMen: 0.02; Rossman Store Sales: 0.00), indicating that the synthetic data does not pose a privacy risk to real individuals. Results indicate that the method can generate accurate synthetic relational data whilst preserving privacy. The model compatability metric value is significantly smaller for the Rossman data (especially the F1-score). This indicates that the number of secondary tables has a bigger impact on the results than the volume of the data and the complexity of its datatypes. Nonetheless, the metric value is sufficiently small for the synthetic data to be of use in testing, analytic and machine learning contexts.

## 5 Conclusion

We propose a novel deep learning method for the generation of realistic synthetic relational data based on graph neural networks and variational autoencoders. By treating links between tables in a relational database as edges in a graph, message passing layers enable information to propagate through the database. Subsequently, the encoder can infer inter-table patterns, which are reproduced during synthesis. Computational experiments on publicly available datasets indicate that our method creates highly realistic synthetic relational data that does not disclose sensitive information of individuals in the original database.

The proposed method works well, even for a database with a large data volume and complex data types. The quantity of tables in the relational database may render synthesis more problematic,

though the results were promising also for a set with two secondary tables. Improved hyperparameter optimization may improve the results further. Future research should inspect whether there is a correlation between the number of tables in the database and the performance of graphVAE. Dedicated heuristics for hyperparameter optimization should also be explored.

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
