# OpenReview forum: "Generating Realistic Synthetic Relational Data through Graph Variational Autoencoders"
_NeurIPS.cc/2022/Workshop/SyntheticData4ML — Neurips 2022 SyntheticData4ML_

### Official Review · Reviewer_TSsF · 2022-10-12
**A novel and interesting graph variational autoencoder based technique**

**Rating:** 7
**Confidence:** 4

**Review:**

In this paper, authors have proposed a novel method for the generation of realistic synthetic relational data by combining graph neural networks and variational autoencoders. The idea looks interesting which is evaluated on two public datasets and shows good results. So, I recommend this work, although, I have following points to further improve the work.
- From the Introduction, it appears there is no work on generating relational data. However, that does not seem the case, as I can notice one reference related to relational data genearation in the references ([20]). How does this method improves over the existing ones? The literature gap is not clear.
- Since each row act as a vertex and some tables have large number of rows which could lead to huge computational cost requiremnts. So, authors need to discuss the computational overhead of the proposed method.
- Authors motivated their study by talking about the privacy concerns so they should discuss some metrics for the same also. I understand there is limited space. Maybe use something like Choi et al (Generating Multi-label Discrete Patient Records using Generative Adversarial Networks) can be discussed.
- I think, in equation (2) the order of two subequations should be reversed. Please check!
- Equation (4) is not clear, please check.

---

### Official Review · Reviewer_ohbi · 2022-10-17
**The authors combine VAEs and GNNs to generate relational data. They show that training classifiers on the generated data leads to similar results obtained by training classifiers on the original training data.**

**Rating:** 7
**Confidence:** 3

**Review:**

It is a well-written paper which was easy to follow for me. There are some minor issues though:
page 1
line 17 1100 -> 1,100
line 18 indicates -> indicate

page 2
Is ∅ used for missing data? If so, please clarify it.

page 3
line 82, 99, 104 So that -> so that, And -> and, Where -> where
line 107 entered -> fed

commas/periods missing after equations (general comment)

page 4
please explain what the target features are.

please report how the manual tuning of hyperparameters are done. did you use a separate validation set for this?

did you use any normalization for numerical values?

did you use one-hot encoding for categorical variables?

there's only one column of date type. perhaps it would make sense to discard that column

references
[12] ... bayes ... -> ... Bayes ...
[14] ... gans ... -> ... GANs ...
[16] .... beta-vae ... -> [12] .... beta-VAE ...

add links and access dates for [21] and [22] please

My main concern is the lack of comparison with alternative approaches, if exists. I am also slightly concerned about the manual tuning of the hyper-parameters.

I thought that this is overall an interesting work. It would have been more useful if one can also generate anomalous data given that tabular data is often messy.

---

### Official Review · Reviewer_Vvtz · 2022-10-17
**Lack of the novelty of the method and concern about the reliability by author’s model to synthesis tabular dataset with inconvincible experiments**

**Rating:** 3
**Confidence:** 4

**Review:**

The author proposes a method that combines the VAE and GNN to generate the new tabular and relational dataset. However, there are several issues:

First of all, the experiment by classification can’t evaluate the generation quality well. The author trained two models where one on training dataset and one on the synthetic dataset, however, the evaluation metrics number from both models are similar isn't direct evidence to show that the structure and data distribution between the two datasets are similar. And the experiment didn’t show any image to illustrate the final generation result.

Secondly, for the tabular datasets, the structure variance, design diversity, etc, are the key challenges to synthesizing a good enough table dataset.  However, few of those key factors are discussed and well considered in modeling or experiments.

---

### Meta-Review · Area_Chair_ydGe · 2022-10-19

**Recommendation:** Accept